# Oleogel Formulations for the Topical Delivery of Betulin and Lupeol in Skin Injuries—Preparation, Physicochemical Characterization, and Pharmaco-Toxicological Evaluation

**DOI:** 10.3390/molecules26144174

**Published:** 2021-07-09

**Authors:** Ramona Daniela Pârvănescu (Pană), Claudia-Geanina Watz, Elena-Alina Moacă, Lavinia Vlaia, Iasmina Marcovici, Ioana Gabriela Macașoi, Florin Borcan, Ioana Olariu, Georgeta Coneac, George-Andrei Drăghici, Zorin Crăiniceanu, Daniela Flondor (Ionescu), Alexandra Enache, Cristina Adriana Dehelean

**Affiliations:** 1Department VIII—Neuroscience, Discipline of Medical Deontology, Bioethics, Faculty of Medicine, “Victor Babes” University of Medicine and Pharmacy, 2nd Eftimie Murgu Square, RO-300041 Timisoara, Romania; ramona.parvanescu@umft.ro; 2Department of Pharmaceutical Physics, Faculty of Pharmacy, “Victor Babeș” University of Medicine and Pharmacy Timisoara, 2nd Eftimie Murgu Square, RO-300041 Timisoara, Romania; farcas.claudia@umft.ro; 3Research Centre for Pharmaco-Toxicological Evaluation, “Victor Babeș” University of Medicine and Pharmacy, 2nd Eftimie Murgu Square, RO-300041 Timișoara, Romania; iasmina.marcovici@umft.ro (I.M.); macasoi.ioana@umft.ro (I.G.M.); draghici.george-andrei@umft.ro (G.-A.D.); ionescu.daniela@umft.ro (D.F.); cadehelean@umft.ro (C.A.D.); 4Department of Toxicology and Drug Industry, Faculty of Pharmacy, “Victor Babeș” University of Medicine and Pharmacy Timisoara, 2nd Eftimie Murgu Square, RO-300041 Timisoara, Romania; 5Department II—Pharmaceutical Technology, Faculty of Pharmacy, “Victor Babes” University of Medicine and Pharmacy, 2nd Eftimie Murgu Square, RO-300041 Timisoara, Romania; olariu.ioana@umft.ro (I.O.); coneac.georgeta@umft.ro (G.C.); 6Formulation and Technology of Drugs Research Center, “Victor Babes” University of Medicine and Pharmacy, 2nd Eftimie Murgu Square, RO-300041 Timisoara, Romania; 7Department of Analytical Chemistry, Faculty of Pharmacy, “Victor Babeș” University of Medicine and Pharmacy, 2nd Eftimie Murgu Square, RO-300041 Timișoara, Romania; fborcan@umft.ro; 8Department of Plastic and Reconstructive Surgery, Faculty of Medicine, “Victor Babeș” University of Medicine and Pharmacy, 2nd Eftimie Murgu Square, RO-300041 Timișoara, Romania; zcrainiceanu@gmail.com; 9Department VIII—Neuroscience, Discipline of Forensic Medicine, Bioethics, Deontology and Medical Law, Faculty of Medicine, “Victor Babes” University of Medicine and Pharmacy, 2nd Eftimie Murgu Square, RO-300041 Timisoara, Romania; esanda2000@yahoo.com

**Keywords:** betulin oleogels, lupeol oleogels, SJL mice, UV radiation, epidermolysis bullosa, skin parameters, cell viability, angiogenesis, CAM assay

## Abstract

The skin integrity is essential due to its pivotal role as a biological barrier against external noxious factors. Pentacyclic triterpenes stand as valuable plant-derived natural compounds in the treatment of skin injuries due to their anti-inflammatory, antioxidant, antimicrobial, and healing properties. Consequently, the primary aim of the current investigation was the development as well as the physicochemical and pharmaco-toxicological characterization of betulin- and lupeol-based oleogels (Bet OG and Lup OG) for topical application in skin injuries. The results revealed suitable pH as well as organoleptic, rheological, and textural properties. The penetration and permeation of Bet and Lup oleogels through porcine ear skin as well as the retention of both oleogels in the skin were demonstrated through ex vivo studies. In vitro, Bet OG and Lup OG showed good biocompatibility on HaCaT human immortalized cells. Moreover, Bet OG exerted a potent wound-healing property by stimulating the migration of the HaCaT cells. The in ovo results demonstrated the non-irritative potential of the developed formulations. Additionally, the undertaken in vivo investigation indicated a positive effect of oleogels treatment on skin parameters by increasing skin hydration and decreasing erythema. In conclusion, oleogel formulations are ideal for the local delivery of betulin and lupeol in skin disorders.

## 1. Introduction

The skin is the largest organ in the human body, acting as a shield for the human organism against the environment and microorganisms. The skin integrity for survival is essential so it is constantly renewed to ensure other functions such as immune surveillance and tactile sensitivity. It is worth noting that humans can endure various internal diseases and can even live with them for a long time, but they cannot live with external diseases or even with a partial absence of epithelia in any part of the body, which represent a huge medical problem [1]. Depending on the etiology, skin wounds can be acute or chronic. The most common skin wounds are acute, resulting from a trauma (blunt, incision, excision) and/or from a burn. After skin injuries occurs, the affected tissue has the ability to stimulate and coordinate the physiological response described in four phases: hemostatic, inflammation, cellular proliferation, and remodeling of the extracellular matrix [2,3]. In the final phase, also called the healing phase, the fibroblasts produce an extracellular matrix, which begins to mature [4], leading to an increase of the mechanical strengthening of the tissue [5]. The wound-healing process is considered closed only after the death of myofibroblasts, vascular cells, and macrophages [6,7,8]. When the wound healing is delayed or not healing at all, then the process becomes chronic, resulting in the disorder of the original tissue structure and homeostasis, with a series of complications such as infections, excessive inflammation, blistering, ulcerations, hypertrophic scars and keloids, pain, itchiness, hyperkeratosis, and so on, which leads to a skin fragility [3,9,10,11]. In the chronic wounds, there is an excessive degradation of the extracellular matrix, and the proliferative phase with tissue regeneration followed by the remodeling phase are absent [6,8].

Skin fragility is a term proposed recently [12] that refers to those primary manifestation at the skin, which is reflected in the body organs or systems [13]. The skin fragility comprises various disorders (e.g., peeling skin disorders, erosive disorders, hyperkeratotic disorders), as well as other disorders of the connective tissue, such as epidermolysis bullosa (EB). The EB, also known as butterfly disease, is a rare genetic disease with skin blistering, due to structural anomalies of the skin, more precisely defects in proteins or the lack of certain anchor proteins. The clinical manifestation of this heterogeneous genetic disorder may include peeling, erosions, ulcerations, itch, wounds, hyperkeratosis, and scars from a minor mechanical stress that disrupts the dermoepidermal junction [13]. To date, there are four known main clinical and genetic features of classical types of EB, namely: EB simplex (EBS), junctional EB (JEB), dystrophic EB (DEB), and Kindler EB (KEB) [14]. These classical forms of EB differ from each other by the level of the blistering within the dermal–epidermal junction and can be distinguished by transmission electron microscopy (TEM), immunofluorescence mapping, or molecular genetic diagnosis, upon morphological analysis of a skin biopsy [15]. It is very important to determine with precision the EB subtype, this aspect being useful in prognostication improvement to facilitate the genetic counseling and prenatal diagnosis. If the exact clinical phenotype and how mutations of the same gene are inherited autosomal dominantly or recessively is determined, then the results can be included in clinical trials and in precision medicine [16,17].

Other disorders that could lead to skin fragility (e.g., erythema, edema, sunburn cell, hyperplastic response, photoaging, or skin cancer) are due to chronic solar UV radiation [18,19]. The UV radiations are considered a major factor in skin cancer or other skin injuries. The most harmful radiation is UVA (320–400 nm), which is absorbed by the deeper layers of the dermis. Much of the UVB radiation (280–320 nm) penetrates the surface layers of the epidermis, reaching at most the upper papillary dermis [20,21]. The UVC radiation (200–280 nm) is the least harmful radiation due to the fact that it is blocked by the ozone layer of the Earth [22]. In the first phase, the UV radiation affects mainly keratinocytes, resulting in an inflammatory effect in which the UVB radiation destroys the internal microstructure of the skin, resulting in its faster aging [23]. Regarding the fibroblasts, the UVB irradiation leads to a significant increase of reactive oxygen species (ROS), DNA damage, and mitochondrial impairment, which leads to a decrease in cell proliferation, apoptosis, and skin fibrosis [24,25].

According to the report of a meeting at the British Association of Dermatologists [26], there are different types of skin fragility, including EB, with no effective pharmacological or genetic treatment option until now. Compared to other skin disease, the EB can be treated by genetic intervention, but this treatment cannot be considered a cure for the disease due to the large number of mutant genes involved and the large area of skin affected by fragility. Precisely for this reason, some manufactured treatments designed to relief the pain and itch and finally to accelerate the wound healing are much appreciated and urgently needed. Tremendous scientific developments have been made in recent years regarding skin injuries, both curative and symptom-relief therapies, and as concerns the EB, most research efforts focus on therapies for severe forms of EB [27,28,29]. Topical therapies have been tested for EB based on evidence from in vitro studies, and publications have reported that a topical agent with anti-inflammatory effect, based on betulin derived from birch bark, has enhanced keratinocyte differentiation [30]. Another literature study reports on the topical therapy with betulin-based oleogel for the treatment of dystrophic EB patients, which was presented in a small open, blindly evaluated phase II pilot study. It was observed that the patients responded in a positive way after the application of oleogel in terms of re-epithelialization and wound healing [31]. In addition, in a phase III clinical trial, another oleogel with betulin was tested to observe its effect on the healing of burn wounds. It was noted that patients treated with betulin oleogel showed a higher cure rate and avoidance of complications such as wound infection [32]. Regarding the effect of lupeol, Pereira-Bessera et al. observed that in vitro, lupeol has a beneficial effect on wound healing due to the fact that it has anti-inflammatory properties [33]. Subsequently, the same team of researchers proved the protective and repairing effect of lupeol in wounds, this time using a murine model [34].

Due to their relatively low toxicity, the properties of pentacyclic triterpenes are of great interest in treating skin diseases, in particular for their wound-healing properties [35].

The present study aims at the (i) preparation and physicochemical characterization of three types of oleogels (blank-oleogel—without active compound, betulin-oleogel, and lupeol-oleogel); (ii) in vitro pharmaco-toxicological investigation regarding the biocompatibility of oleogels on a skin healthy cell line (immortalized human keratinocytes—HaCaT); (iii) in vitro assessment of the potency of oleogels on wound healing and skin re-epithelialization; (iv) drawing of the in ovo irritative profile and (v) examination of the variation of skin parameters following the topical application of oleogels on SJL mice. 

The novelty of the present study consists in the preparation of semi-solid formulations (betulin and lupeol oleogels), with 0.3% content of active compound, which is much lower than in the formulations already present in the literature that are used in clinical trials (10% triterpene dry extract from birch bark). In addition, the prepared oleogel formulations are applied to the UV radiation damaged skin of mice, and the results obtained confirm the enormous efficacy potential of the prepared oleogels, besides the strongest wound-healing activity in vitro, which is another novelty of the study. 

## 2. Results

### 2.1. Physicochemical Properties of Oleogels

#### 2.1.1. Macroscopic Examination 

The organoleptic properties of the prepared oleogels are presented in Table 1 and the appearance of the three oleogels when the physicochemical parameters and biological evaluation were determined are depicted in Figure 1. 

#### 2.1.2. Determination of pH

The pH values obtained for control (Blank OG) and Lup OG formulations were slightly alkaline (Table 1), which was most probably due to the presence of glycerol diester (glyceryl dibehenate) used as a gelling agent for the oily phase. On the other hand, the BET-OG formulation produced a lower pH value (very close to neutral), which can be attributed to the acidic character of the active compound (betulin). 

#### 2.1.3. Rheological Characterization

The results of the steady-state flow (viscosimetric) test are illustrated in Figure 2 as flow and viscosity curves, and they are listed in Table 2 as viscosity and thixotropy values. Table 2 also presents the results of the analysis of the viscometric data by their fitting with the rheological model.

The profile of rheograms revealed that the studied oleogel formulations behave as non-Newtonian pseudoplastic systems with a thixotropic character, as indicated by the hysteresis loop of different areas observable on rheograms (Figure 2). A proportional increase of the share stress with the share rate and the viscosity decrease with an increasing share rate describe the pseudoplastic or shear-thinning behavior, which is a relevant property for topical semi-solid medications. The thixotropy of the studied oleogels is evidenced by the lower values of the shear stress for the same shear rate on the down-ramp, in comparison with up-ramp (Figure 2). Due to these rheological properties, the product will flow easily when it is taken out of the tube (high shear rates are applied), facilitating its local application; in turn, when it is spread on the site of administration (low shear rates are applied), the semi-solid product returns to a higher consistency, remaining at the site of administration. 

Compared to blank oleogel, the viscosity of Bet OG was about 1.4 times higher, whereas the Lup OG showed a very close viscosity value (Table 2). The viscosity values ranged in the specific field of topical semi-solid medications. Comparing the medicated oleogels with control in terms of thixotropy, the Bet OG formulation exhibited a value that was 1.9 times higher, while the Lup OG formulation was 1.5 times less thixotropic (Table 2). The Herschel-Bulkley rheological model fitted all rheograms accurately, the *R* values being higher than 0.9 and indicated that all oleogels behaved as pseudoplastic systems, since the *n* values were lower than 1 (Table 2). Moreover, it can be observed that the flow index values calculated for the blank OG and Bet OG formulations were very close and that of the Lup OG formulation increased slightly, indicating its slightly less shear-thinning behavior. The consistency index *K* value calculated for Bet OG was 1.25 times higher than that of the control, while the Lup OG was 2.35 times less consistent than the control formulation.

#### 2.1.4. Textural Measurements

In the spreadability test, the higher firmness and spreadability values were produced by the blank OG formulation (Table 3), indicating that the highest force is required to spread it—that is, it has the lowest spreadability. Instead, for the Lup OG, the lowest firmness and spreadability values were obtained (Table 3), which reveals it has the easiest spreadability. The oleogel formulation containing betulin developed a 1.5 times higher firmness value and a double work of shear value compared to those obtained for Lup OG, indicating a slightly lower spreadability. When compared to blank formulation, the Bet OG formulation was significantly more spreadable (*p* < 0.05), as the measured firmness and spreadability values were about nine times lower.

#### 2.1.5. FT-IR Investigations of Oleogels

The functional groups of both active compounds (betulin and lupeol) contained in oleogels type formulations were identified by employing the Fourier-Transform Infrared Spectroscopy (FT-IR) analysis.

Figure 3 shows the peaks recorded by a specific wavenumber in the case of Bet-OG and Lup-OG. In order to highlight the presence of active substances in the prepared formulations, spectra of both pure biological substances (betulin and lupeol) were also recorded.

Figure 3A exhibits the FT-IR spectra of betulin and Bet OG, which are rather similar, exhibiting the same main band positions and in some cases relative intensities. The FT-IR spectrum of pure betulin presents relevant absorbption peaks in the high wavenumber region (3479.58, 3381.21, 3224.98, 3076.46, 2945.30, and 2868.15 cm^−1^ (Figure 3A, black line)). These peaks are attributed to OH, CH_3_, and CH_2_ asymmetric and symmetric stretching vibrations, respectively. There are two relevant absorption peaks, at 2357.01 and 2330.01 cm^−1^, which are attributed to C–H stretching vibrations. The dominant peaks were also recorded on the Bet OG spectrum (Figure 3A, red line), at 2918.30, 2850.79, 2358.94, and 2331.94 cm^−1^. The intense peak at 1747.51 is assigned to the carbony (C=O) functional group, which is prominent in the bark spectrum. In the fingerprint region (430–1650 cm^−1^), the FT-IR spectrum of pure betulin presents dominant absorption peaks at 1651.07, 1645.28, 1454.33, 1373.32, 1105.21, 1008.77, 877.61, and 667.37 cm^−1^ together with many other peaks of medium to weak intensity. The peak recorded at 1651.07 cm^−1^ can be assigned to C=C stretching and CH_2_ bending vibrations in the terminal methyl group; this peak is present in both spectra. All the peaks recorded in this spectral range are assigned to the bending vibration of OH, CH_2_, and CH_3_ groups. The peaks at 1373.32, 1008.77, and 983.70 cm^−1^ are attributed to C–O stretching or CH_2_–OH groups. On the Bet OG spectrum, the functional groups are recorded at 1379.10, 1101.35, and at 1062.78 cm^−1^. The intense peak recorded at 877.61 cm^−1^ on the pure betulin spectrum is assigned to the wagging vibration of the CH_2_ in the alkene group. The rest of the medium and weak peaks are assigned to the CH_2_, CH_3_, CH, or OH bonds, and they are also highlighted on the Bet OG spectrum. 

Figure 3B shows the FT-IR spectra of pure lupeol and Lup OG. In this case, the two spectra are very similar. The Lup OG FT-IR spectrum (Figure 3B) had intense absorption peaks at 2922.16, 2850.79, 1747.51, 1471.69, 1180.44, and 717.52 cm^−1^. The intense peaks at 2922.16 and 2850.79 cm^−1^ are assigned to the vibrational stretching of methylene and methyl groups. The peak at 1747.51 cm^−1^ is assigned to the conjugated carbonyl (C=O) functional group. The peaks recorded in the fingerprint region (450–1650 cm^−1^), some relevant and others of medium to weak intensity, are assigned to CH_3_ and CH_2_ bending vibration groups (at 1471.69 and 1379.10 cm^−1^), C–C stretching vibration, OH bending vibration, CH_2_ torsion vibration, and CH bending vibration groups recorded at 1180.44 cm^−1^, as well as C–O groups from CH_2_–OH stretching vibration, CH bending vibration, and CH_3_ and CH_2_ rocking vibration (1103.28 cm^−1^). 

### 2.2. Skin Permeation Study

#### 2.2.1. Ex Vivo Permeation Test

To estimate the bioavailability of a new topical formulation, the ex vivo drug permeation through skin is used as an alternative to in vivo studies in humans. For the skin permeation tests, pig ear skin was selected, as it is an appropriate model membrane to human skin, due to their histological and biochemical similarity. Furthermore, using porcine ear skin, the lack of human skin for research purposes has been overcome [36,37]. The results of betulin and lupeol ex vivo permeation from oleogel formulations through pig ear skin are depicted in Figure 4 and summarized in Table 4. 

As shown in Figure 4, the oleogel vehicle released slightly different maximum amounts of active compound after 16 h of testing, namely 35.08 ± 1.49% corresponding to 185.83 ± 2.77 µg/cm^2^ of betulin and 32.93 ± 2.15% corresponding to 174.45 ± 3.27 µg/cm^2^ of lupeol. The transfer of betulin per unit surface area of membrane in time was slightly lower than that of lupeol (13.38 ± 1.05 µg/cm^2^/h versus 15.57 ± 1.18 µg/cm^2^/h), but without a lag time, which was about 5 h in case of lupeol (Table 4).

#### 2.2.2. Evaluation of Drug Retention on Skin

The amounts of betulin and lupeol accumulated into the skin membrane used in the ex vivo permeation experiments are presented in Table 4. It can be observed that a 3-fold higher amount of active compound (betulin or lupeol) was retained in the skin than that permeated across it.

### 2.3. Cellular Viability Assessment on Human Immortalized Keratinocytes (HaCaT)

In order to evaluate the impact of the three formulations on the viability of HaCaT cells, the Alamar Blue assay has been performed at two time intervals: 24 and 72 h. After 24 h, a slight decrease in the cell viability has been noticed following the treatment with Blank OG (125, 250, and 500 µg/mL). In the case of Bet OG and Lup OG, the cell viability percentages vary in a dose-dependent manner. Thus, Bet OG reduces the viability of HaCaT cells at low concentrations, while at the highest concentration of 500 µg/mL, a stimulatory effect has been remarked (the cell viability percentage is over 100%). On the other hand, Lup OG induced a gradual reduction in cell viability, the most significant value being noted at 500 µg/mL (Figure 5A). A similar trend has been observed after the 72 h stimulation period for all samples (Figure 5B), with the mention that at 24 h, Bet OG and Lup OG exerted the strongest effect at the highest concentration.

### 2.4. Wound Healing or Scratch Assay

In order to verify if Blank OG, Bet OG, and Lup OG interfere with the migration of HaCaT cells, the wound healing (Scratch) assay was employed. Two concentrations (125 and 500 µg/mL) were selected for each sample. The highest wound-healing rate was registered after the cells’ treatment with Bet OG (at 125 µg/mL—93.19%; at 500 µg/mL—100%), followed by Lup OG 125 µg/mL (82.60%) and Control (82.43%). Blank OG, DMSO 500 µg/mL, and Lup OG 500 µg/mL exerted a potent anti-migratory property with wound healing rates of 67.84, 68.70, and 70.70%, respectively (Figure 6).

### 2.5. Hen Embryo Chorioallantoic Membrane Assay (HET-CAM)

The potential toxic and irritating effect of oleogels (OG) was evaluated using the in ovo method, which involves applying samples to the chorioallantoic membrane. The toxic effect was followed in the case of oleogel with betulin (Bet OG) and lupeol (Lup OG), as well as in the case of blank oleogel (Blank OG)—without active compound. The results obtained were reported to negative control (distilled water—H_2_O) and positive control (sodium dodecyl sulfate—SDS 0.5%). To accurately verify the potential toxic effect of oleogels, the solvent used to prepare them, DMSO 0.05%, was also tested.

The three effects observed in the blood vessels (hemorrhage, lysis, and coagulation) were obviously noticed in the case of positive control (SDS) and in the case of solvent (DMSO). In the case of SDS, the highest irritation score was obtained. After applying a volume of 500 µL of SDS, a micro-hemorrhage is observed, which is accompanied by lysis and vascular coagulation. In addition, in the case of SDS, specimen death was registered after less than 24 h. In the case of the solvent, DMSO, it has a lower irritation score than SDS, but higher than in the case of oleogels. After applying a volume of 500 µL of DMSO, a slight coagulation and vascular lysis occurred, which was accompanied by localized hemorrhage (Figure 7). 

In the case of the three oleogels tested, even in the highest concentration, no major changes are recorded in the blood vessels. Thus, we can say that the three oleogels do not have an irritating effect in ovo. In addition, embryo specimens showed a survival rate of over 2 days. Figure 8 shows graphically the values of the irritation score for both positive and negative control, as well as for oleogels and the solvent used for their preparation. From Table 5, it can be seen that the highest irritation score was obtained after the application of SDS, while the lowest values of the irritation score were obtained in the case of oleogels.

### 2.6. Skin Biophysical Parameters Assessment

In order to evaluate the anti-inflammatory effect of Lup and Bet oleogels as treatment for different skin homeostatic parameters changes, an in vivo experiment was performed, using female SLJ as a common model for this aim. The comparative evolution of skin biophysical parameters can be observed in Figure 9. Important differences between the four mice groups were observed in the evolution of skin parameters in the last 20 days and can be reported as positive aspects: an increase of the transepidermal water loss was found in the case of the mice used as control and those who were treated with blank OG (Figure 9A), while decreased values of this parameter were recorded in the groups of mice treated with Bet OG and Lup OG. 

Figure 9B presents the erythema level—the main skin parameter, which can be used to predict the safety of the oleogels, with or without active compounds. Compared to the results obtained for the first parameter, in this case, relevant differences were found between the groups treated with blank OG and those treated with Bet OG and Lup OG. The erythema index presented a noticeable decrease when the dorsal skin of the mice was treated with both oleogels, which contained active compounds, especially when Lup OG was used. The melanin, a natural pigment involved in the skin and hair coloring mechanisms, depicted in Figure 9C, has increased very much in the first 10 days due to the exposure to UV radiation; its index remained at a high level because the oleogels do not modify the skin pigmentation; however, a slight decrease was observed due to the health beneficial effects of the tested samples. Figure 9D shows the evolution of skin hydration following the application of blank OG, Bet OG, and Lup OG. From the analysis of this parameter, we can predict the beneficial potential of the oleogels with Bet and Lup. There is an observed significant increase in skin hydration when mice were treated after UV exposure with both oleogels (Bet OG and Lup OG), as against the mice groups of control and those treated with blank OG.

## 3. Discussion

The present research study was conducted to evaluate the safety and efficacy of three types of oleogel from which two types contain a specific amount of active substances (betulin and lupeol). The first part of the study describes the preparation and characterization of oleogels, with or without active compounds. The obtained physicochemical results are correlated with the ex vivo skin permeation study, in vitro biocompatibility and wound healing property, in ovo anti-irritative potential, and in vivo effect on the evolution of physiological skin parameters. 

Pentacyclic triterpenes are well known to possess a wide range of biological properties including anti-inflammatory, anticancer, antiseptic, antiviral, antimicrobial, and even anti-HIV activities [38,39,40,41,42,43,44]. Betulin and betulinic acid are the main representants of the pentacyclic triterpenes family, and an impressive number of studies were dedicated to these natural compounds due to their multiple pharmacological effects: anti-inflammatory, antiviral, hepatoprotective, antiangiogenic, and antitumoral effects [45]. Recent studies demonstrated the efficacy of these compounds in different diseases, including multiple sclerosis [46], a severe necrotizing herpes zoster in an immunosuppressed patient who had not responded to a conventional topical treatment [47], and actinic keratosis [48].

The composition of the oleogel as a vehicle for betulin and lupeol was selected based on the results of the previous published studies [31,49,50,51]. Betulin and lupeol were incorporated in an oleogel vehicle based on a vegetable oils mixture of sunflower oil and olive oil in a 2:1 ratio. Compritol ATO 888 (glyceryl dibehenate) was used as an organogelator. The selection of oleogel vehicle based on vegetable oils mixture meets current demands for biocompatible and more environmentally friendly semi-solid topical formulations in the pharmaceutical domain [52]. Glyceryl dibehenate, a biocompatible pharmaceutically approved excipient with GRAS status, is frequently used as an oil gelator due to its pronounced lipophilic character and its moderate melting point of 64–74 °C [53]. Furthermore, it has the ability to efficiently entrap active substances in its solid matrix formed in oleogels, consequently delaying their release [54]. 

The organoleptic properties of the obtained oleogels (Table 1) were found to be appropriate for skin application. For all three tested formulations, the obtained pH values were in accordance with the values indicated in the national pharmacopoeia [55] for semi-solids, and they can be tolerated by the skin. In order to explain the differences among the tested oleogels in terms of viscosity, thixotropy, and consistency, the results of previously published studies should be considered, which demonstrated the ability of the triterpene dry extract from the outer bark of birch to act as a gelling agent when suspended in oils [56,57]. Betulin and lupeol (active compounds of the tested oleogels) are components of triterpene dry extract from birch bark, and it can be suggested that due to their particular chemical structure, they differently modify the structural arrangement of the three-dimensional gel network formed by the main gelling agent (glyceryl dibehenate). This results in variations regarding the viscosity, thixotropy, and consistency of the oleogels studied in the present work. Moreover, it can be observed that betulin oleogel is more efficient as a secondary gelling agent compared to lupeol.

Based on the spreadability test results, it can be suggested that the triterpene active components affect the structural arrangement of the three-dimensional network formed by glyceryl dibehenate, which is the main gelling agent of the studied oleogels.

FT-IR spectra, a qualitative analysis, is considered a chemical fingerprint employed to identify the functional groups of organic compounds contained in the prepared oleogels. Analyzing the FT-IR spectra of pure betulin and lupeol, it can be concluded that both triterpenes give identical characteristic peaks. Moreover, the FT-IR spectra recorded almost cannot differentiate between botulin and lupeol. The peaks recorded at 1008.77 and 1043.49 cm^−1^ reflect the triterpene contribution, betulin, and lupeol. The similarities with the spectra of pure active compounds, Bet OG and Lup OG, are remarkable. On the FT-IR spectra of both oleogels, these peaks are recorded at weak intensity. Nevertheless, there are intense highlighted peaks on the FT-IR spectra of Bet OG and Lup OG, which suggest the presence of active compounds in oleogel type formulation, such as CH, CH_2_, and CH_3_ stretching vibration recorded at 2918.30, 1471.69, 1423.47, and 1379.10 cm^−1^ (Bet OG) and at 2922.16, 1471.69, and 1379.10 cm^−1^ (Lup OG); CH_2_ bending vibration, C=C stretching vibration, and C–C–H bending vibration recorded at 1651.07 cm^−1^ (Bet OG); as well as C–C stretching vibration, OH bending vibration, CH_2_ torsion vibration, and CH bending vibration recorded at 1180.44 cm^−1^ on the FT-IR spectra of both Bet OG and Lup OG. These results suggest that FT-IR analysis allows differentiating betulin and lupeol as well as the oleogels type formulations prepared, in spite of their high structural similarity. Our results are in concordance with those of Silverstein et al. [58], Muktar et al. [59], Cînta-Pînzaru et al. [60], and Fălămaș et al. [61].

The ex vivo permeation results revealed that the formulation presenting slower diffusion did not show lag time, as would have been expected. Thus, one can assume that in the case of experimental oleogels containing betulin or lupeol, the calculated lag time depends on the release of active compound from the oleogel but also on its diffusion through the skin membrane. In the first process, the formulation viscosity has an important contribution: the Bet OG formulation being more viscous than the Lup OG formulation (Table 2), the flux and release rate values of betulin were lower than those of lupeol. As the skin thickness was unchanged throughout the permeation experiments, the flux of betulin and lupeol through the membrane is influenced by their lipophilicity correlated with their chemical structure, but also by the diffusion coefficient, which depends on the skin structure. The relatively small values of the Bet and Lup permeation parameters through intact pig ear skin could be attributed to the presence of the stratum corneum, which is the major barrier resisting active compound skin permeation. However, the high retention in the skin of both triterpenes can be attributed to their high affinity to the lipophilic stratum corneum. 

Since biocompatibility is mandatory for topical formulations, the impact of Bet- and Lup-loaded oleogels on the viability of HaCaT cells has been assessed. This particular cell line has been selected as an in vitro experimental model for skin healthy cells based on the fact that keratinocytes (i) represent the most dominant cellular component of the multi-layered epidermis [62]; (ii) stand as defensive shields against external harmful factors (e.g., UV radiation) [63]; and (iii) play a vital role in epidermal wound healing and renewal through a process known as epithelialization, as well as via exerting immune functions [62,64]. Our viability results indicate a lack of cytotoxicity of Blank OG, Bet OG, and Lup OG and a suitable in vitro biocompatibility on HaCaT cells. 

Another important aspect that has been investigated in the present study is the efficacy of Bet OG and Lup OG in skin recovery and renewal. As highlighted by the scratch assay results, the effects induced by the oleogels on the migration of HaCaT cells are dose-dependent, Bet OG being associated with a stimulatory effect by inducing a wound closure rate of 100% at 500 µg/mL, while Lup OG exhibited an anti-migratory activity at the same concentration with a wound-healing rate of 70.70%.

As part of the safety evaluation of the three prepared oleogels (Blank, Bet, and Lup), we used the hen embryo chorioallantoic membrane (HET-CAM) assay, which is a versatile toxicological method for verifying the possible toxic and irritating effect that a substance can have on a biological membrane. Testing the oleogels on the chorioallantoic membrane revealed their lack of irritative potential, the irritation scores obtained for the Bet- and Lup-containing oleogels being less than 1. The use of phytocompounds under various formulations for wound treatment has been extensively documented in the literature, with many articles focusing on the anti-irritant and wound repair effects of pentacyclic triterpenes [43]. 

The impact on different physiological skin parameters was also assessed by non-invasive methods using SLJ mice. The in vivo results obtained in the present study indicate the benefits of the oleogels with Bet and Lup at the skin level, which are characterized by reduced values of erythema and enhanced values of skin moisture (both parameters are significantly modified compared to the group of mice control and the group treated with Blank OG—without active compound). Important changes of skin parameters were obtained in the first 10 days due to the exposure to UV radiation: transepidermal water loss, erythema, and melanin levels have grown a lot, while the skin hydration has dropped. Erythema, skin hydration, and transepidermal water loss are the main parameters that can be associated with some skin pathological aspects, including skin injury, inflammation, or in some cases, infection [65].

## 4. Materials and Methods

### 4.1. Materials

#### 4.1.1. Chemical and Reagents

Betulin, lupeol, Dulbecco’s Modified Eagle Medium (DMEM), and sunflower and olive oils were purchased from Sigma-Aldrich (Darmstadt, Germany). Compritol 888 ATO (glyceryl dibehenate) was received from Gattefossé (Saint-Priest, France) as a free sample. All other materials were of analytical purity and were used as received.

#### 4.1.2. In Vitro Evaluations

The in vitro evaluations were performed using the HaCaT cell line, which was ac-quired from ATCC (LGC Standards GmbH, Wesel, Germany) as a frozen vial. The cells were cultured in their specific growth medium (DMEM) supplemented with FCS (10%) and penicillin–streptomycin mixture (1%), and maintained in a humidified atmosphere (37 °C, and 5% CO_2_) during the experiments.

For the in vitro testing, the oleogels were dissolved in DMSO until a stock solution of 50 mg/mL was obtained. The stock solution was further used to prepare the four different concentrations of oleogels (62.5, 125, 250, and 500 µg/mL) for the stimulation of the HaCaT cell line.

### 4.2. Animals

In order to evaluate the effect of Lup and Bet oleogels on skin, an in vivo experiment was performed using eight female SLJ (10 weeks, weight = 18 ± 2 g) mice that were purchased from Charles River (Sulzfeld, Germany). The mice were kept in plastic cages, in the same conditions, in accordance with the Guide for the Care and Use of Laboratory Animals (1996, published by National Academy Press): 12 h/12 h light/dark cycle, at a normal temperature (24 ± 1 °C) and humidity above 55%, fed ad libitum, and had free access to water. There was a 10-day adaptation period, after which the mice were divided in 4 equal groups: group control (no intervention), group blank (mice treated with blank oleogel), group Lup (mice treated with Lup oleogel), and group Bet (mice treated with Bet oleogel). The in vivo experiment was performed according to the following protocol: each oleogel was topically applied (500 µL) on their dorsal skin every two days.

The experiment has been approved by the Bioethical Committee of “Victor Babes,” University of Medicine and Pharmacy Timisoara (protocol code no. 3/30.10.2020), and it respected the international regulations (European Directive 2010/63/EU and the national law 43/2014).

### 4.3. Methods

#### 4.3.1. Preparation of Betulin/Lupeol Oleogels and Blank Formulations

To prepare the oleogels, accurately weighed amounts of Compritol 888 ATO (15% *w*/*w*) and active compound (0.3% *w*/*w*) were dissolved in the mixture of sunflower oil/olive oil (2:1) by heating at 45 °C and stirring continuously at 600 rpm until a clear, homogenous oil solution was obtained. Then, the solution was cooled down at room temperature to form the oleogel by gelation. Blank (control) oleogel formulation without active compound was also prepared by the same procedure.

#### 4.3.2. Physicochemical Characterization of Oleogels

##### pH Measurement

The pH of blank and medicated oleogel formulations was measured at 25 ± 2 °C by the potentiometric method described in pharmacopoeia [66], using a pH-meter (Sension™ 1 portable digital pH meter, Hach Company, Columbus, OH, USA). One g of oleogel was dispersed in 20 mL of distilled water by heating at 45 °C and stirring for 1 min. Then, the obtained dispersion was cooled down at room temperature and filtered. The pH of the filtrate was measured in triplicate.

##### Rheological Characterization

Static rheometry was used to determine the flow behavior and viscosity of the plain and medicated oleogels. Rheological tests were performed in triplicate, at the temperature of 23 °C, using a RheoStress 1 stress-controlled rheometer (Thermo Haake, Warnford, Hampshire, UK) equipped with a plate/plate (PP60Ti, plate diameter 60 mm) geometry. In steady-state flow experiments, oleogels were subjected to a shear rate ramp-up and ramp-down (0.05–100 1/s), and the corresponding shear stress (Pa) and viscosity (Pa·s) were recorded and plotted versus shear rate to obtain the rheograms and viscosity curves respectively. Mathematical modeling of the rheograms and viscosity curves was performed by means of the rheometer software (RheoWin 4 version 4.3, Thermo Fisher Scientific, Waltham, MA, USA), using the Herschel-Bulkley model (Equations (1) and (2)):(1)$$\tau \;=\;\tau_{0}\;+\;K\;\cdot \;{\dot{\gamma }}^{n}$$
(2)η = τ0γ˙ + K ⋅ γ˙n − 1
where τ is the shear stress (Pa), $\tau_{0}$ is the yield stress (Pa), η is the apparent viscosity (Pa·s), *K* is the consistency index (Pa·s^n^), $\dot{\gamma }$ is the shear rate (1/s), and *n* is a rheological exponent called flow behavior index (dimensionless). For 0 < *n* < 1, the system exhibits a shear thinning behavior; usually, the smaller the value of *n* is, the more the system is shear thinning.

##### Textural Measurements

Two important textural parameters (firmness and spreadability) of oleogel samples were measured using a texture analyzer (TA.XT Plus, Stable Micro Systems, London, UK), equipped with a load cell of 5 kg. The textural experiments were performed using a specific accessory (TTC Spreadability Rig HDP/SR, Stable Micro Systems, London, UK), consisting of a male (positive) and five female (negative) acrylic 90° cones, which were precisely matched. The female cones were filled with oleogel samples using a spatula and avoiding air incorporation; then, the surface was evened. During the test, after the male cone was lowered over a distance of 23 mm at a speed of 3 mm/s to the sample surface, it penetrated the sample, forcing it to flow outward at 45° between the cones’ surfaces. From the force versus time curve, the maximum force and work of shear (the area under the curve for positive force region) were selected as indicators of oleogel firmness and spreadability, respectively. Lower force values required to displace the sample from the female cone indicate easier spreadability. Four samples of each oleogel formulation were tested at room temperature. Data analysis was performed using the Exponent software ver. 6.1.18.0 (Stable Micro Systems, London, UK).

##### FT-IR Spectroscopy

The FT-IR spectra of Bet OG, Lup OG, as well as pure betulin and lupeol were recorded in the range from 4000 to 400 cm^−1^ on KBr pellets, under reduced pressure. A Shimadzu Prestige-21 spectrometer (Duisburg, Germany) operating with a peak resolution of 4 cm^−1^, at room temperature conditions (24 °C), was employed to perform all the spectra.

#### 4.3.3. Ex Vivo Skin Permeation Studies

##### Preparation of the Skin Samples

The ex vivo drug permeation experiments were carried out using porcine ear skin ex-cised from 4-month-old, female and male domestic pigs slaughtered in a local abattoir (Timisoara, Romania). The ears were washed with cold tap water immediately after excision; then, their dorsal region was clipped of bristles. To obtain the skin samples, the skin from the dorsal ear region was verified to not present any lesions or spots and was dermatomed (Electrodermatome Acculan 3 Ti, Aesculap-aBBraun Company, Center Valley, PA, USA) to a thickness of 500 µm. The skin samples were used immediately in permeation experiments. The integrity of the skin was evaluated by visual examination for physical damage, excluding unsuitable samples.

##### Ex Vivo Permeation Tests

Permeation tests were performed using six vertical diffusion cells (Microette-Hanson system, 57-6AS9 model, Hanson, Chatsworth, CA, USA), with an effective permeation area of 1.767 cm^2^ and a receptor volume of 6.5 mL. To ensure sink conditions, the receptor compartment was filled with freshly prepared phosphate-buffered saline solution (PBS), pH = 7.4, with 10% hydroxypropyl-β-cyclodextrin [67]. Prior to each permeation test, skin samples were kept in a receptor medium for 0.5 h at ambient temperature. The skin was carefully mounted between the donor and receptor compartments of the vertical diffusion cells, with stratum corneum up. About 300 mg of oleogel formulation was weighed into each donor compartment, which was then fitted to the diffusion cell. The receptor compartment content was continuously stirred at 600 rpm, and the diffusion cells were maintained at 32 ± 1 °C throughout the experiment. Samples of 0.5 mL receptor fluid were withdrawn at predetermined time intervals over 24 h (1, 2, 3, 4, 5, 6, 7, 8, 9, 10, 11, 12, 13, 14, 15, 16, 17, 18, 20, 22, and 24 h) and replaced with an equivalent volume of fresh receiver medium to maintain a constant volume. The collected samples were analyzed for Bet and Lup oleogels content by the UV spectrophotometric method, at 207 nm wavelength. The assay was linear in the Bet and Lup oleogels concentration range of 0.8–8 μg/mL (*y* = 0.101*x*, *R*^2^ = 0.999) and respectively 0.4–4 μg/mL (*y* = 0.218*x* − 0.001, *R*^2^ = 0.999). Three replicates of each experiment were performed.

##### Evaluation of Drug Retention in the Skin

To evaluate the amount of betulin and lupeol accumulated into the skin, drug ex-traction was performed according to the method described by Günther et al. with minor modification [68]. Briefly, after completing the permeation test, the two compartments of each vertical diffusion cell were disassembled, and the remnant oleogel was removed from the skin surface, which was washed three times with a 0.5% *w*/*w* sodium lauryl sulfate aqueous solution to rinse off the skin. The piece corresponding to the permeation surface was cut from the skin samples, dried at ambient temperature, and afterwards weighed and cut into smaller pieces, which were transferred in glass vials containing 2 mL methanol. After 24 h of extraction at 4 °C, the skin dispersion in methanol was homogenized (Silent Crusher M homogenizer, Heildolph, Germany) for 3 min and centrifuged for 5 min at 3500 rpm. The drug content in the supernatant was analyzed by UV spectrophotometry at 207 nm wavelength, after appropriate dilution with methanol. Skin drug retention, which was considered as the amount of drug in the supernatant, was expressed in µg/cm^2^ of skin sample.

##### Data Analysis of Ex Vivo Drug Permeation Studies

A cumulative amount of Bet and Lup oleogels permeated through the model membrane (µg/cm^2^), which was plotted as a function of time (hour). The permeation rate of the substance at steady state (flux, *J_s_*, µg/cm^2^/h) and the lag time (*t_L_*, h) were calculated from the slope and the x intercept of the linear portion of the plots of cumulative amount of active compound permeated versus time in steady-state conditions, respectively. The release rate (*k*) values were calculated using the pseudo steady-state slopes from plots of cumulative amount of active compound permeated through membrane (µg/cm^2^) vs. square root of time.

#### 4.3.4. Cell Viability Assessment 

The viability of the cells was assessed by the means of the Alamar Blue technique. In brief, the cells were cultured in 96-well plates (1 × 10^4^ cells/200 µL/well) and treated with increasing concentrations (62.5, 125, 250, and 500 µg/mL) of Blank OG, Bet OG, and Lup OG for 24 and 72 h, respectively. At the end of the treatment, 20 µL of Alamar Blue reagent were added in each well, followed by the absorbance measurements at 570 and 600 nm using a xMark™ Microplate Spectrophotometer (BioRad Laboratories, Inc., Thermo Fischer Scientific, Carlsbad, CA, USA). The cellular viability percentages were calculated according to the formula described in a previous study [69].

#### 4.3.5. Wound Healing or Scratch Assay

The impact of oleogels on the migration of HaCaT cells was evaluated by applying the wound-healing (Scratch) assay. In this regard, the cells were seeded in 12-well plates at a density of 10^5^ cells/well and allowed to attach. When the proper confluence was reached, a manual scratch was performed in the middle of each well using a 10 µL sterile tip, which was followed by washing the cells with PBS and treatment with Blank OG, Bet OG, and Lup OG (125 and 500 µg/mL). Additionally, the cells were stimulated with DMSO 500 µg/mL to verify the influence of the solvent on their migratory potential. Representative images (at 10× magnification) of the scratch area were recorded at 0 and 24 h using an Olympus IX73 inverted microscope (Olympus, Tokyo, Japan). The initial and final wound widths were measured using the CellSense Dimension 1.17 software (Olympus, Tokyo, Japan), and the wound-healing rates were calculated by applying a formula previously described by Felice et al. [70].

#### 4.3.6. Hen Embryo Chorioallantoic Membrane Assay

Fertilized eggs (*Gallus gallus domesticus*) were provided by a local farmer from Timisoara. They were disinfected with 70% alcohol, dated, and then placed in the incubator at a constant temperature of 37 °C and constant humidity. On the 3rd day of incubation, 8–9 mL of albumen was extracted in order to allow the chorioallantoic membrane to detach from the inner shell of the egg, so that the blood vessels were easier to observe. On the 4th day of incubation, a window was cut at the top of the egg, which was then covered with adhesive tape, and the eggs were put back in the incubator until the beginning of the experiment.

To evaluate the biocompatibility and toxicity of oleogels, the Hen’s Egg Test (HET-CAM) method was applied. Three eggs were used for each oleogel. A concentration of 500 µg/mL from each oleogel was tested; the highest concentration was tested in vitro. For the negative control, distilled water was used, and for the positive control, 0.5% sodium dodecyl sulfate (SDS) solution in H_2_O was used. The solvent used to prepare the oleogels, DMSO, was also tested. The changes observed into CAM were evaluated using a stereomicroscope (Discovery 8 Stereomicroscope, Zeiss, Göttingen, Germany), and the images were performed (Axio CAM 105 color, Zeiss) before and after a 5 min application. All images were processed using ImageJ v 1.50e software (U.S. National Institutes of Health, Bethesda, MD, USA, https://imagej.nih.gov/ij/index.html, accessed on 14 March 2021). The effects followed for five minutes in the blood vessels were as follows: H, vessel lysis—*L,* and coagulation and extra vascular—*C*. The analytical method used to determine the irritant effect was to calculate the irritation score (IS) using the formula (Equation (3)) [71]:(3)$$IS\;=\;5\;\times \;\frac{301\;-\;H}{300}\;+\;7\;\times \;\frac{301\;-\;L}{300}\;+\;9\;\times \;\frac{301\;-\;C}{300}.$$

#### 4.3.7. Skin Biophysical Parameters Assessment

The mice were divided into 4 equal groups after the first 10 days, while they were exposed to UV radiation: group blank (exposed to UV and no other intervention), group control (mice exposed to UV and treated with the base of oleogel), group Lup (UV and respectively Lup oleogel treatment), and group Bet (UV and respectively Bet oleogel treatment). The UV exposure was described in a previous paper of our research group [72]: the cages were placed under a VL-6.M/6W (312 nm wavelength and 680 µW/cm^2^ intensity at 15 cm) tubes (Vilber Lourmat, Collégien, France) and the irradiation was daily, 5 min/day, in the first 10 days of the experiment when every mouse has received a total dose around 200 J/m^2^ UV radiation. In the last 20 days of the experiment, the mice were separated into the four groups, and they received different topical applications. 

The measurements of the skin parameters were carried out with a Multiprobe Adapter from Courage-Khazaka electronic GmbH (Köln, Germany). A professional Mexameter^®^ MX 18 probe was used to record the melanin and erythema levels, while the transepidermal water loss (TWL) and skin hydration were measured using the probes Tewameter^®^ TM 300 and Corneometer^®^ CM 825, respectively; the evaluations were performed by the same operator 30 min after every treatment; there were used differences (Δ parameter) between an instant measure and an initial one for every mouse.

### 4.4. Statistical Analysis

All measurements were performed as triplicate, and the continuous variables are presented as mean and standard error. Regarding the in vitro assays, the statistical differences between means were compared by applying the one-way ANOVA analysis followed by Tukey’s and Dunnett’s multiple-comparisons post-tests (** *p* < 0.01; *** *p* < 0.001; **** *p* < 0.0001).

Regarding the skin parameters, two-way ANOVA analysis was applied to determine the statistical differences, which is followed by a Bonferroni post-test (* *p* < 0.05; ** *p* < 0.01; *** *p* < 0.001).

## 5. Conclusions

Due to the myriad of therapeutic effects exerted by triterpenes (e.g., anti-inflammatory, antimicrobial, anti-angiogenic, anti-tumor), two oleogels containing betulin and lupeol as active ingredients were prepared and characterized from a physicochemical point of view. The obtained results revealed that the developed oleogel formulations presented adequate features in terms of pH, and organoleptic, rheological (flow behavior, viscosity, thixotropy), and textural (firmness and spreadability) properties. The ex vivo permeation studies revealed the ability of oleogel formulations to assure the penetration and permeation of betulin and lupeol through porcine ear skin. Moreover, the retention of both triterpenes in the skin was demonstrated. In addition, Bet OG and Lup OG exerted a good biocompatibility in vitro, as well as a lack of toxicity in ovo and in vivo. Regarding their efficacy, the oleogels positively influenced the skin parameters in mice by reducing erythema and increasing skin moisture. Regarding the potential effect on skin re-epithelialization, Bet OG exerted the strongest wound-healing activity in vitro.

## Figures and Tables

**Figure 1 molecules-26-04174-f001:**
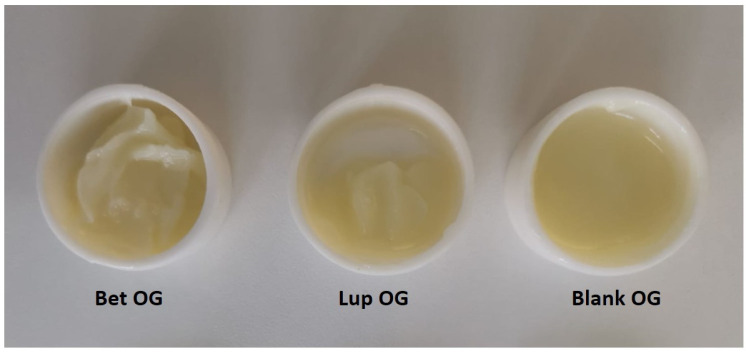
The appearance of the three oleogels.

**Figure 2 molecules-26-04174-f002:**
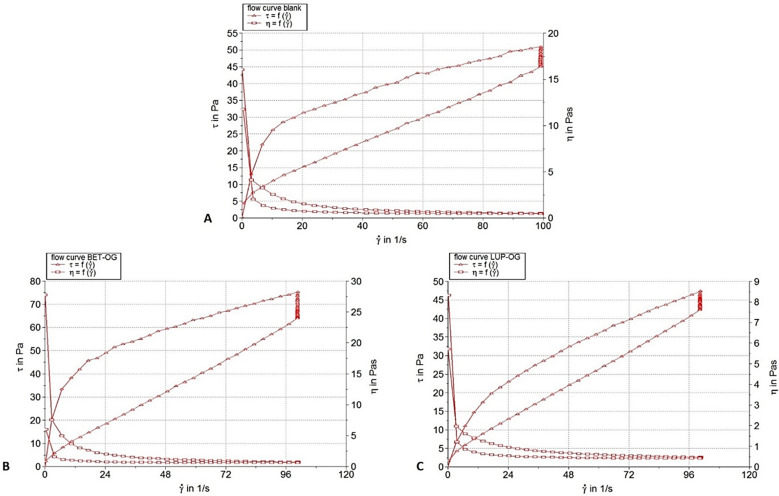
Flow and viscosity curves of the studied oleogels: (**A**)-Blank OG, (**B**)-Bet OG, and (**C**)-Lup OG.

**Figure 3 molecules-26-04174-f003:**
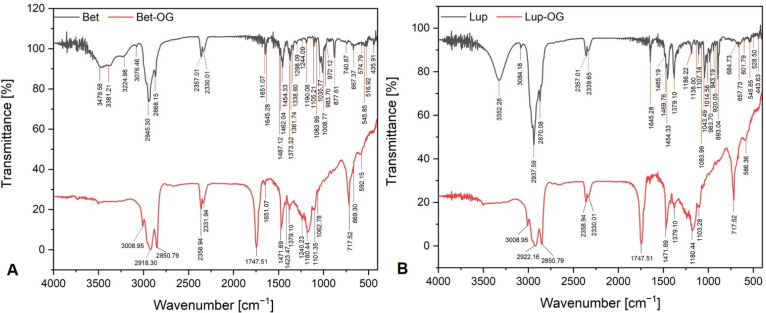
FT-IR spectra of Bet OG (**A**-red line) and Lup OG (**B**-red line) alongside with FT-IR spectra of pure botulin (**A**-black line) and lupeol (**B**-black line).

**Figure 4 molecules-26-04174-f004:**
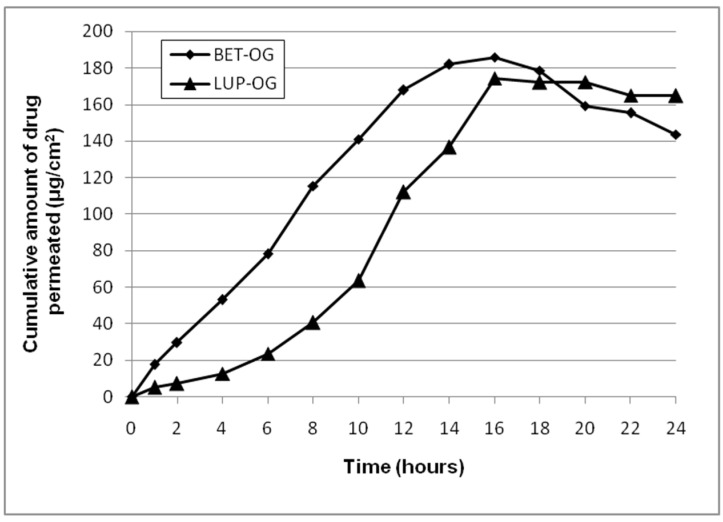
Ex vivo betulin and lupeol permeation profiles through pig ear skin from oleogel formulations.

**Figure 5 molecules-26-04174-f005:**
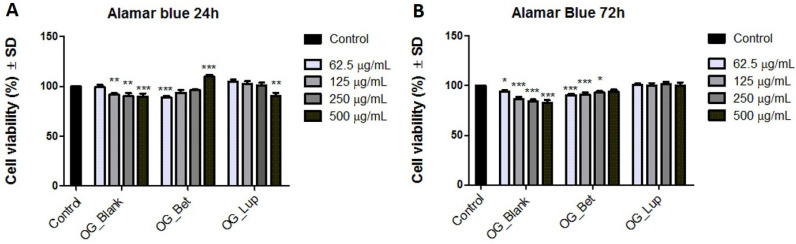
Cell viability assessment of the three oleogels (Blank OG, Bet OG, and Lup OG) in HaCaT cells at 24 h (**A**) and at 72 h (**B**) post-stimulation by means of Alamar blue assay. The results are expressed as a cell viability percentage (%) normalized to control cells stimulated with cell medium. The data represent the mean values ± SD of three independent experiments. A one-way ANOVA analysis was applied to determine the statistical differences compared with control-treated cells followed (* *p* < 0.1; ** *p* < 0.01; *** *p* < 0.001).

**Figure 6 molecules-26-04174-f006:**
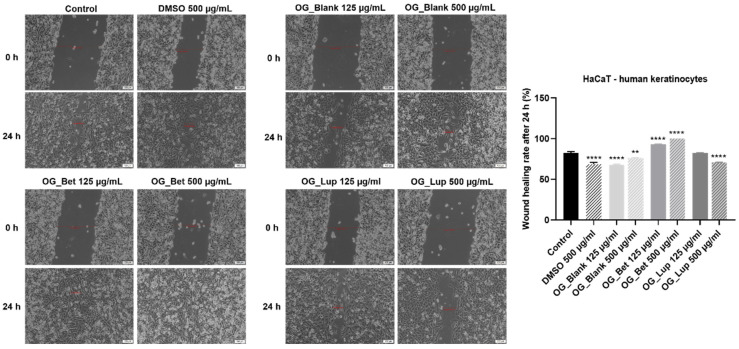
The influence of Blank OG, Bet OG, and Lup OG (125 and 500 µg/mL) on the migratory ability of human immortalized keratinocytes (HaCaT) by applying the wound-healing assay. The bar graphs are presented as wound closure percentage after 24 h of treatment in comparison to the initial surface at 0 h. The results are expressed as mean values ± SD of three experiments performed in triplicate. The statistical differences between treated and non-treated (Control) cells were identified by performing the one-way ANOVA analysis and the Dunnett’s multiple comparisons post-test (** *p* < 0.01; **** *p* < 0.0001). The scale bars represent 100 µm.

**Figure 7 molecules-26-04174-f007:**
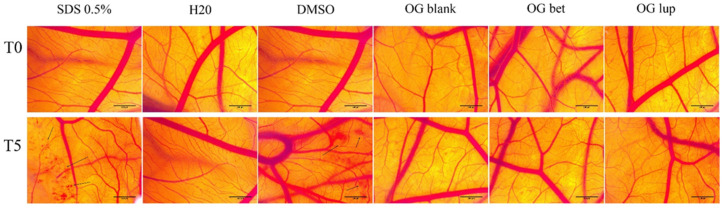
Analysis of the irritant potential of Blank OG, Bet OG, and Lup OG by the HET-CAM method. Stereomicroscopic images of CAMs inoculated with negative control—H_2_O, positive control—SDS, solvent used to solubilize OGs—DMSO 0.05% and test compounds 500 µg/mL. The black arrow indicates areas with massive bleeding in the case of SDS and DMSO.

**Figure 8 molecules-26-04174-f008:**
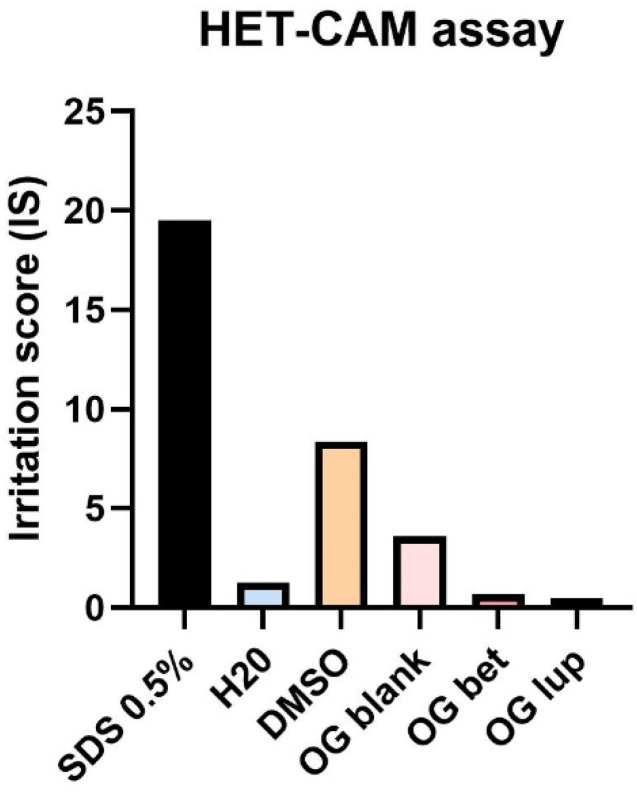
Graphical representation of the irritation score obtained after applying the samples on the chorioallantoic membrane. The positive control is represented by SDS, and the negative control is represented by distilled water.

**Figure 9 molecules-26-04174-f009:**
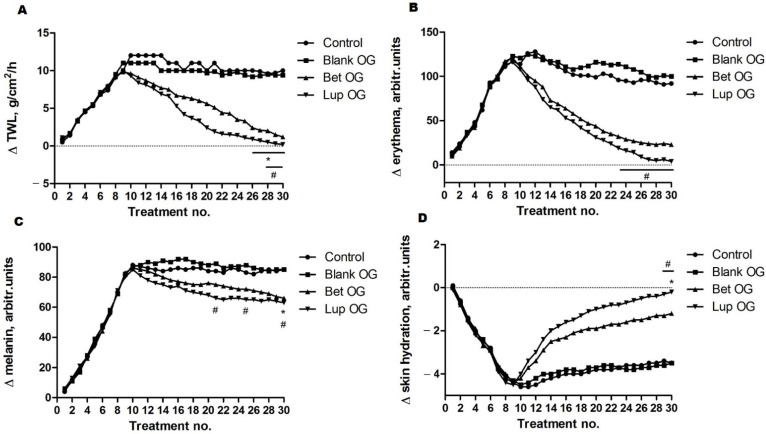
Comparative evolution of skin biophysical parameters: (**A**) Transepidermal water loss (for Lup and Bet OG vs. blank OG # *p* < 0.05; and for Lup, Bet, and blank OG vs. control * *p* < 0.05); (**B**) Erythema (for Lup and Bet OG vs. blank OG # *p* < 0.05); (**C**) Melanin (for Lup and Bet OG vs. blank OG # *p* < 0.05; and for Lup, Bet, and blank OG vs. control * *p* < 0.05); (**D**) Skin hydration (for Lup and Bet OG vs. blank OG # *p* < 0.05; and for Lup, Bet, and blank OG vs. control * *p* < 0.05). The statistical differences were determined using a two-way ANOVA analysis followed by a Bonferroni post-test.

**Table 1 molecules-26-04174-t001:** Organoleptic parameters and pH values of the as-synthesized experimental oleogels.

Formulation	Organoleptic Parameters				pH
	Appearance	Color	Odor	Opacity	
Blank OG	smooth-oily	yellowish-white	odorless	opaque	7.803 ± 0.025
Bet OG	smooth-oily	yellowish-white	odorless	opaque	7.117 ± 0.005
Lup OG	smooth-oily	yellowish-white	odorless	opaque	7.750 ± 0.01

**Table 2 molecules-26-04174-t002:** Rheological parameters (viscosity and thixotropy), parameters of the Herschel-Bulkley model describing the flow curve (*K* and *n*) and Pearson’s correlation coefficient (*R*).

Formulation	Blank OG	Bet OG	Lup OG
Viscosity (Pa·s)	0.478	0.681	0.442
Thixotropy (Pa/s)	1314	2451	901.1
Parameters of Herschel–Bulkley model *K* (Pa·s^n^) *n* (dimensionless)	4.428 0.503	5.55 0.535	1.882 0.679
*R* (flow curve) *R* (viscosity curve)	0.9107 0.9907	0.9005 0.9734	0.9621 0.9811

**Table 3 molecules-26-04174-t003:** Textural parameters (firmness and spreadability) of studied oleogels.

Formulation	Firmness (g)	Spreadability (g.s)
Blank OG	273.80 ± 0.824	165.09 ± 0.603
Bet OG	30.53 ± 0.258	18.23 ± 0.397
Lup OG	20.37 ± 0.147	9.02 ± 0.105

**Table 4 molecules-26-04174-t004:** The permeation and release parameters of active compounds from oleogel formulations through pig ear skin ^a^ and the amounts of active compounds accumulated in the skin samples.

Formulation	Permeation Parameters		Release Parameter	Drug Retention in the Skin (μg/cm^2^ of Skin)
	*J_s_* (μg/cm^2^/h)	*t_L_* (h)	*k* (μg/cm^2^/h^1/2^)	
Bet OG	13.38 ± 1.05	-	69.60 ± 2.38	580.62 ± 9.31
Lup OG	15.57 ± 1.18	5.1	99.76 ± 4.51	539.27 ± 8.18

^a^*J_s_*: steady-state flux; *t_L_*: lag time; *k*: release rate.

**Table 5 molecules-26-04174-t005:** Irritation score (IS) for SDS, DMSO, H_2_O, Blank OG, Bet OG, and Lup OG 500 µg/mL and the occurrence time of hemorrhage (*t_H_*), lysis (*t_L_*), and coagulation (*t_C_*).

	SDS 1%	DMSO 0.5%	H_2_O	Blank OG	Bet OG	Lup OG
**IS**	19.52	8.36	1.27	3.61	0.67	0.46
***tH***	27 s	52	300	300	300	300
***tL***	19 s	168	300	287	300	296
***tC***	22 s	264	260	192	280	290

## Data Availability

Data presented in this study are available on request from the corresponding author.

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
