# Peer review of "Oleogel Formulations for the Topical Delivery of Betulin and Lupeol in Skin Injuries—Preparation, Physicochemical Characterization, and Pharmaco-Toxicological Evaluation"

_molecules, 2021, doi:10.3390/molecules26144174_

Round 1

Reviewer 1 Report

Manuscript “ Oleogel Formulations for the Topical Delivery of Betulin and  Lupeol in Skin Injuries - Preparation, Physicochemical Characterization, and Pharmaco-Toxicological Evaluation” represents a contribution to field of chemistry and all interfacing disciplines.

Text is clear and easy to read. Conclusions are consistent with the evidence and arguments presented. The research topic is relatively original. Before accepting the manuscript, it is essential that the authors:

  • It is necessary to improve the part  2.1. Physicochemical properties of oleogels. It is necessary to add a chemical analysis of oleogels (FT-IR or HPLC). Discuss new results.

Author Response

Thank you very much for your appreciation. We performed FT-IR analysis for oleogels with active compound content, as you recommended. Please check the lines: 211-253; 432-450 and 583-587.

Reviewer 2 Report

The manuscript describes the preparation and characterization of betulin and lupeol based organogels. 

In my opinion, the introduction should be more focused in the triterpenoids and oleogels or other formulations addressed for this type of compounds. The authors should clearly indicated what it is starting point of their work, if the oleogel described has the same composition that one tested in clinical trials or not. It is not clear the novelty of the work, if the oleogel composition, or their characterization.

Moreover, the studies should be completed with ex vivo permeation studies. 

Author Response

Thank you very much for your recommendations. We reviewed everything you indicated. In addition, we performed ex vivo permeability studies. Please check the entire manuscript, especially the lines: 121-132; 143-149; 254-282; 451-465 and 588-643.

Round 2

Reviewer 1 Report

The manuscript is sufficiently improved to publication in Molecules. Accept the manuscript without modification.

Reviewer 2 Report

In my opinion,  the introduction should be focused on the use of terpenoids. The current introduction is too long and not properly addressed

In general, all the manuscript is too long. The authors should try to be more concise.

Add errors in figure 4